# Social Determinants of Stroke Hospitalization and Mortality in United States’ Counties

**DOI:** 10.3390/jcm11144101

**Published:** 2022-07-15

**Authors:** Randhir Sagar Yadav, Durgesh Chaudhary, Venkatesh Avula, Shima Shahjouei, Mahmoud Reza Azarpazhooh, Vida Abedi, Jiang Li, Ramin Zand

**Affiliations:** 1Geisinger Neuroscience Institute, Geisinger Health System, Danville, PA 17822, USA; sagar.randhir@gmail.com (R.S.Y.); dpchaudhary@geisinger.edu (D.C.); sshimashah@gmail.com (S.S.); 2Department of Pediatrics, University of Florida College of Medicine, Jacksonville, FL 32207, USA; 3Department of Neurology, Penn State Health Milton S. Hershey Medical Center, Hershey, PA 17033, USA; 4Department of Molecular and Functional Genomics, Weis Center for Research, Geisinger Health System, Danville, PA 17822, USA; vavula1@geisinger.edu (V.A.); jli@geisinger.edu (J.L.); 5Departments of Clinical Neurological Sciences and Epidemiology, University of Western Ontario, London, ON N6A 3K7, Canada; azarpazhoohr@gmail.com; 6Department of Public Health Sciences, College of Medicine, The Pennsylvania State University, Hershey, PA 17033, USA; vidaabedi@gmail.com; 7Neuroscience Institute, The Pennsylvania State University, Hershey, PA 17033, USA

**Keywords:** stroke outcomes, stroke hospitalization rate, stroke death rate, social determinants, ecological study, education, income, female head of household

## Abstract

(1) Background: Stroke incidence and outcomes are influenced by socioeconomic status. There is a paucity of reported population-level studies regarding these determinants. The goal of this ecological analysis was to determine the county-level associations of social determinants of stroke hospitalization and death rates in the United States. (2) Methods: Publicly available data as of 9 April 2021, for the socioeconomic factors and outcomes, was extracted from the Centers for Disease Control and Prevention. The outcomes of interest were “all stroke hospitalization rates per 1000 Medicare beneficiaries” (SHR) and “all stroke death rates per 100,000 population” (SDR). We used a multivariate binomial generalized linear mixed model after converting the outcomes to binary based on their median values. (3) Results: A total of 3226 counties/county-equivalents of the states and territories in the US were analyzed. Heart disease prevalence (odds ratio, OR = 2.03, *p* < 0.001), blood pressure medication nonadherence (OR = 2.02, *p* < 0.001), age-adjusted obesity (OR = 1.24, *p* = 0.006), presence of hospitals with neurological services (OR = 1.9, *p* < 0.001), and female head of household (OR = 1.32, *p* = 0.021) were associated with high SHR while cost of care per capita for Medicare patients with heart disease (OR = 0.5, *p* < 0.01) and presence of hospitals (OR = 0.69, *p* < 0.025) were associated with low SHR. Median household income (OR = 0.6, *p* < 0.001) and park access (OR = 0.84, *p* = 0.016) were associated with low SDR while no college degree (OR = 1.21, *p* = 0.049) was associated with high SDR. (4) Conclusions: Several socioeconomic factors (e.g., education, income, female head of household) were found to be associated with stroke outcomes. Additional research is needed to investigate intermediate and potentially modifiable factors that can serve as targeted interventions.

## 1. Introduction

The World Health Organization has broadly defined social determinants of health as “the circumstances in which people are born, grow, live, work, and age, and the systems put in place to deal with illness [1].” The American Heart Association has considered socio-economic position (SEP, incorporating wealth and income, education, employment/occupational status, and other factors), race and ethnicity, social support (including social networks), culture (including language), access to medical care, and residential environments as the determinants of cardiovascular diseases [2].

The associations of SEP with stroke incidence and outcomes, such as hospitalization, mortality, and disability in the United States (US), are evident as shown by previous studies (Appendix A) [3,4,5,6,7,8,9,10]. Similar associations have been noted with other determinants, such as access to care, environment, social support, ethnicity, and culture [6,11,12,13]. Various studies have reported the association of stroke incidence and outcome with one or few social determinants. However, there is a paucity of reported population-level studies on the association of stroke hospitalization and mortality with a large set of social determinants. Ecological studies are useful to study large numbers of variables from existing data sets. The goal of this ecological county-level study was to determine the association between social determinants and stroke hospitalization and mortality rates in the US. We hypothesized that social determinants have associations with stroke hospitalization and death rates and variations might exist in the strength and direction in various geographic regions.

## 2. Materials and Methods

We conducted an ecological analysis of social determinants of stroke hospitalization and death rates in 3226 counties/county-equivalents of the states and territories of the US. The study was conducted and reported according to the Strengthening the Reporting of Observational Studies in Epidemiology (STROBE) guidelines.

### 2.1. Study Population and Outcome Measures

This study comprised an analysis of sixteen predictor variables and two outcome variables across all the counties/county-equivalents of the states and territories in the US. The outcomes of interest for this study were all stroke hospitalization rate (SHR) per 1000 Medicare beneficiaries in 2015–2017, and all stroke death rate (SDR) per 100,000 population in 2016–2018. The SDR represented individuals of all ages irrespective of race/ethnicity and gender. On the other hand, SHR represented only Medicare beneficiaries (65+ years) irrespective of race/ethnicity and gender. The median value of the outcomes was used to define two groups of each outcome—those with higher and lower SHR and SDR.

### 2.2. Data Elements

We extracted publicly available data for the predictor and outcome variables from the “Interactive Atlas of Heart Disease and Stroke” website of the Centers for Disease Control and Prevention (CDC) [14]. The data elements available at the CDC were compiled from various sources (https://www.cdc.gov/dhdsp/maps/atlas/data-sources.html, accessed on 9 April 2021). All the data elements, their respective definitions, and data sources are included in Appendix A.

Data elements in this study included county-level variables associated with socioeconomic status (SES), healthcare access, and risk factors. The predictors considered for both SHR and SDR were the presence of hospitals (with and without neurological services) in the county, age-adjusted obesity, age-adjusted physical inactivity, age-adjusted diabetes, income inequality (Gini index), female head of household, no college degree, park access, and food assistance. Median household income, poverty, and unemployment data were only available for the year 2018 and thus these were only considered for SDR and not SHR. Cost of care, heart disease prevalence, and blood pressure medication nonadherence data were only available for Medicare beneficiaries and thus these were only considered for SHR.

### 2.3. Statistical Analysis

Descriptive statistics such as median ± inter-quartile range (IQR) were used for continuous variables and count and percentage were used for categorical variables. Further, the counties were divided into two groups with low or high rates of outcome based on their median value. For comparison between the two groups, Mann–Whitney U-test was used for continuous variables and Pearson’s chi-squared test was used for categorical variables. Pearson’s correlation coefficient was used to calculate the correlation between the variables. For both predictor and outcome variables, values outside of three IQRs were removed as outliers and the variables were scaled and centered. These removed outlier values were treated as missing values. The missingness percentage was calculated for each variable.

To examine the association between the predictors and the outcomes, first, a univariate binomial generalized linear mixed model was used and all predictor variables with *p* < 0.1 were included in the multivariate model. A multivariate binomial generalized linear mixed model was used in which the state of the county was included as a random effect and the median age of the county, population density, and the demographics of the county (percentage of Caucasian, African American, Hispanic/Latino, Asian population) were used as covariates. We applied a multivariate binomial generalized linear mixed model which aims to minimize the false-positive associations due to population or relatedness structure and to increase the power by applying a specific correction to such structures [15]. Variance inflation factor (VIF) was used to examine multicollinearity and variables with VIF > 5 were excluded from the multivariate model.

In addition, standardized hospitalization ratio and standardized mortality ratio were calculated using the national population as the reference (stroke hospitalization rate = 11.7 per 1000 Medicare beneficiaries; stroke death rate = 37.4 per 100,000). The log of standardized hospitalization ratio and standardized mortality ratio were modeled weighted by expected outcome counts for each county.

A *p*-value of less than 0.05 was considered significant for all analyses. All statistical analyses were performed using R version 4.0.3 (R Foundation for Statistical Computing, Vienna, Austria) [16].

## 3. Results

### 3.1. Social Determinants and Their Correlation with Outcomes

The median (IQR) SHR was 11.7 (9.5–13.3) per 1000 Medicare beneficiaries while the median SDR was 39.0 (33.8–44.2) per 100,000. The median household income was found to be USD 51,000 (IQR 44,000–59,000). People living in poverty, food assistance recipients, and the unemployment rate were found to be 14.1% (IQR 10.8–18.3%), 12.9% (IQR 8.6–17.9%), and 3.9% (IQR 3.1–4.9%), respectively. Further, the proportion of families with a female head of the household was 10.6% (IQR 8.4–13.4%). As for education, 80.7% (IQR 74.5–84.9%) did not have four years of college (Table 1).

We observed a median (IQR) rate of 25.9% (22.5–29.6%) for age-adjusted physical inactivity, 33.0% (28.9–36.6%) for obesity, and 10.0% (7.8–12.7%) for diabetes in the counties/county-equivalents of the states and territories in the US. Among the Medicare beneficiaries, the prevalence of heart disease was 35.6% (31.9–39.5), and the cost of care per capita with heart disease at outpatient was USD 3972 (3223–5091). Moreover, only 14.0% (4.0–30.0%) of the US population had access to a park within half a mile proximity (Table 1).

We also found variations in the access to care across the counties/county-equivalents. While 77% of counties/county-equivalents had one or more hospitals, only 28.8% were found to have a hospital with neurological services (Table 1). 

Heart disease among Medicare beneficiaries (r = 0.58), female head of household (r = 0.47), and food assistance (r = 0.45) were the top three socioeconomic factors with the highest positive correlations with SHR, while the cost of care (r = −0.32) and park access (r = −0.26) were negatively correlated with SHR. For SDR, food assistance (r = 0.46), poverty (r = 0.45), and age-adjusted physical inactivity (r = 0.42) were the three factors with the highest positive correlations, while household income (r = −0.38) and park access (r = −0.28) were negatively correlated with SDR. The correlation between other predictors and outcomes is shown in Figure 1A,B. The correlation matrix and heatmap of all predictor variables are given in Appendix A.

### 3.2. Comparison between Counties with Low vs. High Rate of Outcomes

The counties with a higher SHR and SDR had a higher proportion of patients with age-adjusted physical inactivity, age-adjusted obesity, and age-adjusted diabetes (all *p* < 0.001). Similarly, the counties with lower rates of SHR and SDR had a higher percentage of population with access to a park within half-a-mile proximity, and a lower percentage of population without four years of college (all *p* < 0.001). The presence of hospitals in the counties/county-equivalents was associated with lower SHR (*p* < 0.010) and SDR (*p* < 0.068). Further, the presence of hospital(s) with neurological services in the counties/county-equivalents was associated with higher SHR but lower SDR (all *p* < 0.012).

Among Medicare beneficiaries, the counties with higher SHR also had a higher percentage of the population with heart disease, and non-adherence to blood pressure medication while the higher cost of care for heart disease was seen in counties with lower SHR. In the case of death rates, counties with higher SDR had more income inequality, people living in poverty, food assistance recipients, unemployment, uninsured, and households with a female head while those counties with lower SDR had higher household income (Table 1).

### 3.3. Social Determinants Associated with Stroke Hospitalization Rate (SHR)

Except for income inequality, the unadjusted odd ratios (OR) of all independent variables considered for SHR were significant in the univariate analyses (Table 2). In the multivariate binomial generalized linear mixed model, heart disease prevalence (OR = 2.03, 95% CI 1.66–2.49, *p* < 0.001), blood pressure medication nonadherence (OR = 2.02, 95% CI 1.5–2.73, *p* < 0.001), age-adjusted obesity (OR = 1.24, 95% CI 1.06–1.44, *p* = 0.006), presence of hospitals with neurological services (OR = 1.9, 95% CI 1.41–2.57 *p* < 0.001), and female head of household (OR = 1.32, 95% CI 1.04–1.67, *p* = 0.021) were associated with high SHR while cost of care per capita for Medicare patients with heart disease (OR = 0.5, 95% CI 0.42–0.6, *p* < 0.01) and presence of hospital (OR = 0.69, 95% CI 0.5–0.95, *p* < 0.025) were associated with low SHR (Table 2).

All variables which were significant to the multivariate binomial mixed model were also significant in the model for log standardized hospitalization ratio except for age-adjusted obesity and presence of hospital (Appendix A).

### 3.4. Social Determinants Associated with Stroke Death Rate (SDR)

In the univariate analyses, all variables except presence of hospital were shown to have significant unadjusted OR (Table 3). In the multivariate binomial generalized linear mixed model, median household income (OR = 0.69, 95% CI 0.57–0.83, *p* < 0.01) and park access (OR = 0.85, 95% CI 0.74–0.98, *p* = 0.023) were associated with low SDR while no college degree (OR = 1.24, 95% CI 1.03–1.5, *p* = 0.024) was associated with high SDR (Table 3).

All variables which were significant in the multivariate binomial mixed model were also significant in the model for log standardized mortality ratio. In addition, female head of household was also found to be significantly associated with outcome in the model for log standardized mortality ratio (Appendix A).

### 3.5. Variations in Stroke Outcomes and Social Determinants by the US Census Regions

High SHR and SDR were mostly seen in the South census region followed by the eastern part of the Midwest region (Figure 2A). Socioeconomic determinants which were significant in the multivariate models, such as income, education, female head of household, and even park access, show a similar pattern to the stroke outcomes (Figure 2B–E). A map of US counties showing the pattern of the other significant predictors is given in Appendix A.

## 4. Discussion

Our findings have shown that socioeconomic determinants, such as income, education, female head of household, and park access, are significantly associated with stroke outcomes in multivariate analyses. Other significant predictors were prevalence of heart disease, blood pressure medication nonadherence, cost of care, age-adjusted obesity, and presence/absence of hospitals with neurological services.

Prior studies have identified socioeconomic disparities in ischemic and hemorrhagic stroke [10,17,18,19,20]. SES has been noted to have an inverse association with stroke incidence and mortality in the US (Appendix A) while low income was observed to be independently associated with stroke [21]. Similar to the US, the associations of socioeconomic disparities with stroke have been found across other nations showing that socioeconomic disparities increased stroke mortality risk [22,23], stroke mortality [23,24,25], both short-term and long-term stroke outcomes [22,26,27,28], and disability [5,23,29,30]. Findings support the hypothesis of double suffering, i.e., population with lower SES have more long-term illness and experience it with greater intensity and frequency [31]. The observed differences in illness severity can be partly explained by socioeconomic factors such as income [31]. Moreover, the interplay between SES, the actual risk of stroke, stroke risk factors, health access, and health insurance is complex, resulting in different patterns of associations [32].

Lower SEP is highly associated with modifiable risk factors while almost half of the stroke mortalities are due to its poor management. Thus, it is important to consider the high-risk population of lower SEP for targeted stroke prevention efforts [25,28]. Physical activity is a modifiable factor which can be promoted for primary prevention. The number of parks in urban cities is positively associated with physical activity [33], better psychological health, and a cleaner environment while park-based physical activity is negatively associated with park distance [34].

Studies have shown associations of stroke targeted care with better stroke outcomes. At stroke centers, a 2.5% absolute reduction in 30-day mortality was noted [35] while the number needed to treat (NNT) to prevent stroke was found to be 29 versus 40 at general hospitals [36]. In the same manner, stroke units were found to be associated with a lower rate of mortality and decreased length of stay [37,38,39,40]. This study showed that counties/county-equivalents with the presence of hospitals with neurological services had a higher than median SHR but lower than median SDR. Neurological and technical advances in diagnostic modalities, stroke treatment, care, and preventive strategies have a great impact on stroke mortality [41]. Counties/county-equivalents in the US with neurological facilities might have higher SHR due to referral from nearby regions while better diagnosis and treatment could have subsequently reduced the mortality.

There has been a significant geographical variation in the burden of stroke in the US that has persisted over decades [42,43]. In this regard, the Southeastern region of the US is known as the stroke belt owing to its higher stroke mortality rate [42,44,45]. Similarly, stroke incidence [46,47,48] and stroke case-fatality have also shown regional variations [49]. Such findings were consistent with our study. Stroke incidence only partly explains the disparities in stroke mortality, which is further associated with other factors such as racial and geographical disparities [46]. In Canada, the incidence and mortality rate of cardiovascular diseases declined at the national level [50] and province-based level [51,52]; however, these trends were heterogeneous across specific vascular conditions, in terms of magnitude [53] and unexpected peaks, such as a higher rate of stroke admissions for younger Canadians [54]. Some variations can even be seen locally. In Ontario, for example, vascular death varied from 3.2 to 5.7 events per 1000 person-years across Local Health Integration Networks, which may be due to differences in patient characteristics and the health system [55]. Likewise, disparities in health access [56], obesity, physical activity [57], diabetes, and cardiovascular diseases [58] have been reported based on the level of urbanization.

Structural inequities deeply embedded in the fabric of society produce systemic disadvantages faced by one social group and ultimately shape health outcomes [59]. Previous evidence has shown structural racism steadily leads to health inequalities [60]. A recent study has shown an association of structural racism with racial disparities in stroke readmissions [61]. Further, social determinants impact health indirectly through intermediate structural factors such as quality of health care, its access, support service, housing, and transportation facilities [62]. It has been shown that the education level of the head of the household is negatively related to stroke mortality [63]. Female-headed households face the challenges of lower education, poverty, economic insecurity, and health problems [64]. Female-headed households are twice as likely to be poor (27.9 vs. 12.3%), more likely to be food-insecure (30.3 vs. 11.8%), and the median income of female-headed households was significantly lower than the general population according to the 2013 estimates from the U.S. Department of Commerce, U.S. Census Bureau [65,66]. These challenges faced by female-headed households can partially explain the association with stroke.

Strength and limitations: This study considered data of 3226 US counties/county-equivalents of the states and territories in the US and explored a comprehensive number of publicly available socio-determinants for stroke death and hospitalization. The difference in the age range considered for stroke deaths (all ages) and hospitalization (65+ years) was the main limitation of this study. The latter limited our ability to adequately compare findings between death and hospitalization rates. Missingness was also seen in all the variables. It was noted that the overseas US territories had higher rates of missingness while missingness patterns for the mainland counties were deemed to be randomly missing. It should be noted that rates extracted from the source were estimated rates and their precision depends on the number of counts and population size of each county. The dichotomization of the outcomes in this study could distort the association of the variables with the outcomes. The estimated effects when modeling the log standardized hospitalization/mortality ratio were not strong compared to modeling the dichotomized outcomes. The differences in the significance and strength of association of the variables with the outcomes in the two models can be explained partially by the difference in the threshold used in the two models. Various studies have shown social support, residential environment, neighborhood, culture, and language as determinants of stroke which were not evaluated in this study. There were some variations in the study design as well as outcomes and predictors considered by other studies with which our findings were compared. Different variables in this paper represented data recorded over different points/periods from 2015 to 2018. Furthermore, there may be spatial variations in the study variables within the counties. Finally, as with all ecological studies, our study may be prone to “ecological fallacy” which refers to the potential biases that occur when associations that are seen between variables at the aggregate level do not represent the associations between the variables at the individual level [1,67].

## 5. Conclusions

Several socioeconomic factors (e.g., income, education, female head of household, and park access) were found to be associated with stroke outcomes. Additional research is needed to investigate intermediate and potentially modifiable factors that can serve as targeted interventions.

## Figures and Tables

**Figure 1 jcm-11-04101-f001:**
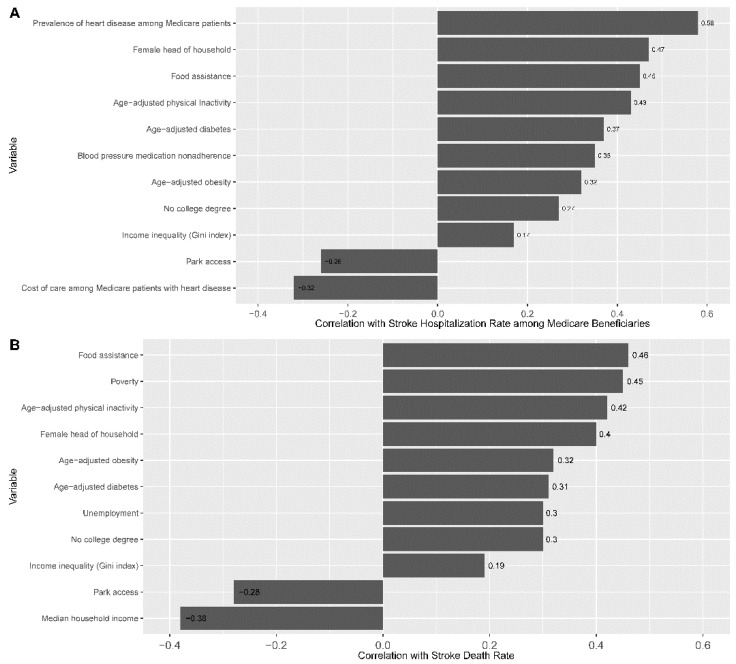
Correlation of continuous variables with outcome variables. (**A**) Correlation of continuous variables with stroke hospitalization rate (SHR) per 1000 Medicare beneficiaries, 65+, all races/ethnicities, both genders; (**B**) correlation of continuous variables with stroke death rate (SDR) per 100,000, all ages, all races/ethnicities, both genders.

**Figure 2 jcm-11-04101-f002:**
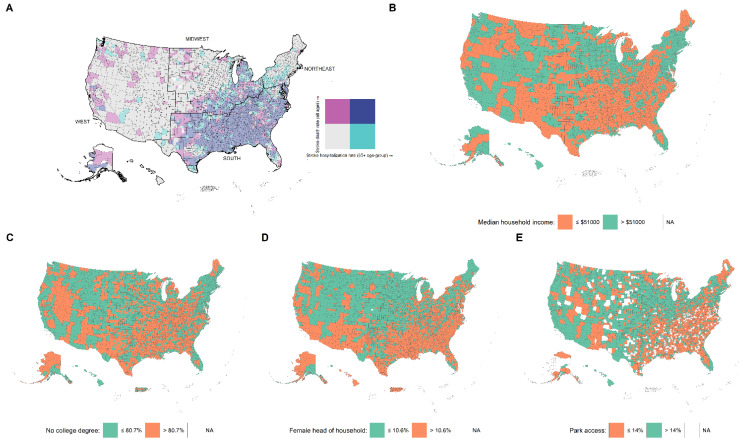
Map of US counties and county-equivalents showing (**A**) stroke hospitalization rate (SHR) among Medicare beneficiaries and stroke death rate (SDR), (**B**) median household income, (**C**) no college degree, (**D**) female head of household, (**E**) park access.

**Table 1 jcm-11-04101-t001:** Comparison between counties/county-equivalents of the states and territories in the US with lower and higher stroke hospitalization rate (SHR) and stroke death rate (SDR).

Data Elements	Unit	Missing %	Overall	Low Hospitalization Rate (≤11.7)	High Hospitalization Rate (>11.7)	*p*-Value	Low Death Rate (≤ 39.0)	High Death Rate (>39.0)	*p*-Value
			Median (IQR)	Median (IQR)	Median (IQR)		Median (IQR)	Median (IQR)	
Heart disease among Medicare beneficiaries	%	0.1	35.6 (31.9, 39.5)	32.7 (29.7, 36.0)	38.5 (35.5, 41.6)	<0.001	-	-	-
Blood pressure medication non-adherence (Medicare Part D beneficiaries)	%	1.8	23.1 (19.8, 26.2)	21.3 (18.2, 24.5)	24.6 (21.5, 27.2)	<0.001	-	-	-
Cost of care (per capita for Medicare beneficiaries diagnosed with heart disease)	USD	0.3	3972.0 (3222.5, 5091.2)	4510.0 (3402.5, 6013.8)	3665.0 (3151.8, 4366.8)	<0.001	-	-	-
Poverty	%	2.6	14.1 (10.8, 18.3)	-	-	-	12.2 (9.6, 15.1)	16.5 (13.1, 20.8)	<0.001
Unemployment	%	0.2	3.9 (3.1, 4.9)	-	-	-	3.6 (2.9, 4.7)	4.1 (3.5, 5.1)	<0.001
Household income	USD	2.6	51,000.0 (44,000.0, 59,000.0)	-	-	-	55,000.0 (48,000.0, 63,000.0)	46,000.0 (41,000.0, 53,000.0)	<0.001
Income inequality (Gini index)	%	0.2	0.4 (0.4, 0.5)	0.4 (0.4, 0.5)	0.4 (0.4, 0.5)	<0.001	0.4 (0.4, 0.5)	0.5 (0.4, 0.5)	<0.001
Food assistance	%	2.6	12.9 (8.6, 17.9)	9.8 (6.6, 14.1)	15.7 (12.2, 20.3)	<0.001	10.2 (6.9, 14.6)	15.5 (11.6, 20.4)	<0.001
Female head of household	%	0.2	10.6 (8.4, 13.4)	9.1 (7.1, 11.5)	12.0 (10.0, 14.7)	<0.001	9.4 (7.5, 11.7)	12.0 (9.6, 14.9)	<0.001
No college degree	%	0.2	80.7 (74.5, 84.9)	78.6 (71.6, 82.5)	83.3 (77.7, 86.6)	<0.001	78.3 (70.7, 82.6)	83.1 (78.2, 86.5)	<0.001
Park access	%	2.7	14.0 (4.0, 30.0)	22.0 (8.0, 38.0)	9.0 (2.0, 21.0)	<0.001	22.0 (8.0, 38.0)	9.0 (2.0, 21.0)	<0.001
Age-adjusted physical inactivity	%	2.6	25.9 (22.5, 29.6)	24.0 (20.6, 27.0)	28.1 (24.8, 31.9)	<0.001	24.0 (20.7, 27.0)	28.1 (24.8, 31.8)	<0.001
Age-adjusted obesity	%	2.6	33.0 (28.9, 36.6)	31.0 (27.0, 34.8)	34.6 (31.3, 37.9)	<0.001	31.5 (27.5, 35.0)	34.4 (30.7, 38.0)	<0.001
Age-adjusted diabetes	%	0.2	10.0 (7.8, 12.7)	8.6 (6.9, 10.8)	11.5 (9.4, 13.9)	<0.001	8.8 (7.1, 11.0)	11.3 (9.0, 13.9)	<0.001
			***n* (%)**	***n* (%)**	***n* (%)**		***n* (%)**	***n* (%)**	
Hospitals with Neurological services	%	0.1	927 (28.8)	441 (27.0)	486 (31.1)	0.012	521 (32.0)	406 (25.6)	<0.001
Hospital present	%	0.1	2482 (77.0)	1296 (79.2)	1179 (75.3)	0.010	1278 (78.5)	1202 (75.8)	0.068

**Table 2 jcm-11-04101-t002:** Univariate and multivariate binomial generalized linear mixed model analyses for stroke hospitalization rate (SHR) per 1000 Medicare beneficiaries, 65+, all races/ethnicities, and both genders.

	Unadjusted OR		Adjusted OR	
Variable	OR with 95% CI	*p*-Value	OR with 95% CI	*p*-Value
Prevalence of heart disease among Medicare patients	2.55 (2.18–2.99)	<0.001	2.03 (1.66–2.49)	<0.001
Blood pressure medication nonadherence	2.88 (2.34–3.54)	<0.001	2.02 (1.5–2.73)	<0.001
Cost of care among Medicare patients with heart disease	0.6 (0.52–0.68)	<0.001	0.5 (0.42–0.6)	<0.001
Hospital with neurological services	1.22 (0.99–1.5)	0.062	1.9 (1.41–2.57)	<0.001
Age-adjusted obesity	1.83 (1.64–2.04)	<0.001	1.24 (1.06–1.44)	0.006
Female head of household	2.01 (1.76–2.29)	<0.001	1.32 (1.04–1.67)	0.021
Hospital present	0.7 (0.56–0.89)	0.003	0.69 (0.5–0.95)	0.025
No college degree	1.39 (1.25–1.54)	<0.001	1.13 (0.94–1.36)	0.189
Park access	0.85 (0.76–0.95)	0.004	0.91 (0.78–1.06)	0.224
Age-adjusted physical inactivity	1.86 (1.64–2.11)	<0.001	1.08 (0.9–1.3)	0.426
Food assistance	2.02 (1.78–2.3)	<0.001	1.09 (0.87–1.37)	0.453
Age-adjusted diabetes	1.47 (1.31–1.64)	<0.001	1 (0.86–1.15)	0.977
Income inequality (Gini index)	1.04 (0.94–1.16)	0.47		

**Table 3 jcm-11-04101-t003:** Univariate and multivariate binomial generalized linear mixed model analyses for stroke death rate (SDR) per 100,000, all ages, all races/ethnicities, and both genders.

	Unadjusted OR		Adjusted OR	
Variable	OR with 95% CI	*p*-Value	OR with 95% CI	*p*-Value
Median household income	0.62 (0.56–0.69)	<0.001	0.69 (0.57–0.83)	<0.001
Park access	0.81 (0.73–0.9)	<0.001	0.85 (0.74–0.98)	0.023
No college degree	1.44 (1.3–1.59)	<0.001	1.24 (1.03–1.5)	0.024
Poverty	1.83 (1.63–2.04)	<0.001	-	-
Age-adjusted diabetes	1.25 (1.13–1.39)	<0.001	0.9 (0.79–1.03)	0.121
Food assistance	1.94 (1.71–2.19)	<0.001	-	-
Age-adjusted physical inactivity	1.71 (1.52–1.92)	<0.001	1.13 (0.95–1.33)	0.174
Unemployment	1.63 (1.44–1.84)	<0.001	1.05 (0.89–1.25)	0.554
Female head of household	1.49 (1.32–1.68)	<0.001	0.98 (0.8–1.19)	0.821
Income inequality (Gini index)	1.17 (1.05–1.29)	0.003	1.04 (0.9–1.2)	0.620
Age-adjusted obesity	1.46 (1.32–1.62)	<0.001	1.03 (0.89–1.19)	0.713
Hospital with neurological services	0.71 (0.58–0.86)	0.001	1.00 (0.76–1.31)	0.985
Hospital present	0.92 (0.74–1.15)	0.471		

## Data Availability

The data used in this study is available at the “Interactive atlas of heart disease and stroke” website of the Centers for Disease Control and Prevention (http://nccd.cdc.gov/DHDSPAtlas, accessed on 9 April 2021).

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
