# Peer review of "Social Determinants of Stroke Hospitalization and Mortality in United States’ Counties"

_jcm, 2022, doi:10.3390/jcm11144101_

Round 1
Reviewer 1 Report
Dear Authors,
thank you for possibility to read and to evaluate your manuscript.
The chosen topic is original and important. There is a very limited scientific knowledge on the Social Determinants of Stroke Hospitalization and Mortality. I find the methods of analysis, including statistics, relevant and appropriate. The literature is up-to-date, and conclusions are based on the original data. The text of manuscript is clear and easy to read.
While appreciating the aims and quality of your work, I still two questions.
First, did you have the possibilities and aims to compare thes ocioeconomic disparities in ischemic and hemorrhagic stroke?
Second, do you have any explanations for how female head of household may be associated with stroke outcomes? If you have any thoughts on this, is it worth mentioning in the discussion?
Thank you.
Author Response
Response to Reviewer 1 Comments
We would also like to thank the reviewer for their time and effort to provide us with valuable feedback on this paper. We have made changes to the manuscript to reflect the suggestions that have been provided. We hope our edits and responses below can address these concerns.
Reviewer 1
Dear Authors,
thank you for possibility to read and to evaluate your manuscript.
The chosen topic is original and important. There is a very limited scientific knowledge on the Social Determinants of Stroke Hospitalization and Mortality. I find the methods of analysis, including statistics, relevant and appropriate. The literature is up-to-date, and conclusions are based on the original data. The text of manuscript is clear and easy to read.
While appreciating the aims and quality of your work, I still two questions.
Point 1: First, did you have the possibilities and aims to compare the socioeconomic disparities in ischemic and hemorrhagic stroke?
Response 1: Thank you for the comment. There was a possibility to compare the socioeconomic disparities in ischemic and hemorrhagic stroke separately (‘CDC Interactive Atlas of Heart Diseases and Stroke’ provides county level data for ischemic and hemorrhagic stroke separately as well). Our preliminary exploration of the data revealed similar patterns in socioeconomic variables for both subtypes of stroke. Thus, it was decided to examine socioeconomic disparities in hospitalization and death rate for stroke as whole (two dependent variables instead of four dependent variables in the case of analyzing ischemic and hemorrhagic stroke separately).
Point 2: Second, do you have any explanations for how female head of household may be associated with stroke outcomes? If you have any thoughts on this, is it worth mentioning in the discussion?
Response 2: We appreciate the comment. It has been shown that the education level of head of household is negatively related to stroke mortality (Khaw et al.). Female headed households face the challenges of lower education, poverty, economic insecurity and health problems (Lafta et al). Female-headed households and twice likely to be poor (27.9% vs. 12.3%), more likely to be food-insecure (30.3% vs. 11.8%) and median income of female-headed household was significantly lower than the general population according to the 2017 estimates from the U.S. Department of Commerce, U.S. Census Bureau. These challenges faced by female-headed households can partially explain the association with stroke. But additional research is needed to explore this association between female head of households and stroke.
We have added the above in the discussion section.
Reviewer 2 Report
The authors reported an interesting study on relationship between socioeconomic status and stroke outcomes. They reported that several socioeconomic factors, such as education, income, female head of house-hold, were associated with stroke outcomes. They focused on the stroke death and hospitalization. However, I don't think this study is novel. There are a lot of evidence on this topic.
Author Response
Response to Reviewer 2 Comments
We would also like to thank the reviewer for their time and effort to provide us with valuable feedback on this paper. We have made changes to the manuscript to reflect the suggestions that have been provided. We hope our edits and responses below can address these concerns.
Reviewer 2
Point 1: The authors reported an interesting study on relationship between socioeconomic status and stroke outcomes. They reported that several socioeconomic factors, such as education, income, female head of house-hold, were associated with stroke outcomes. They focused on the stroke death and hospitalization. However, I don't think this study is novel. There are a lot of evidence on this topic.
Response 2: We would like to thank you for the comment. We agree that there are a few studies which have looked at the relationship between socioeconomic status and stroke outcomes. Previous studies have either analyzed patient level data or were ecological studies limited to smaller geographical size. We believe the strengths of this manuscript lie in the design of the study and size of data (county-level data from all the states and territories of the US). This study also looked at univariate and multivariate associations of a large number of socioeconomic variables with stroke outcomes compared to previous studies. We also reported association of stroke outcomes with variables like park access which was reported previously.

Reviewer 3 Report
This topic is interesting, but several points, mainly in terms of analysis, should be addressed before publication. For details please find the file attached .

Round 2
Reviewer 2 Report
I agree that this study has novelty from county-level data from all states and territories in the US. Therefore, I change my previous opinion and agree to accept this study. Thank you for your efforts.
Author Response
We would like to thank the reviewer for their time and effort to provide us with their valuable feedback on this paper.
Reviewer 3 Report
Thank you for taking the time to revise the manuscript and to consider my comments.
I understand the limitation as to data availability. I saw CDC’s website providing tabulated mortality data, but that may not be the case of the authors’ data.
Although the authors employed my second approach, I do not think they responded the first half of comment #1. It seems natural to me to model SMR, where the exponential of the coefficients represents the SMR ratio by the corresponding variables. In this approach, the estimated effects were not so strong, the reasonable finding that exemplifies the dilution of individual-level associations in ecological studies. In contrast, the main analysis of this study seems excessively complicated in that it models whether the naïve point estimates happened to fall below or above their median, which is itself a random variable specified by the point estimates. The results suggest that the latter approach appears to overestimate the effects of the variables compared with the SMR approach. Although I may be wrong, I suppose that the apparent overestimation may have resulted from picking up random variations because of the randomness of the dichotomic threshold (the random variable median).
Followings are secondary comments—ignorable if you employ only SMR approach.
The authors say, “all variables which were significant to the multivariate binomial mixed model were also significant in the model for log standardized hospitalization ratio with the exception of age-adjusted obesity and presence of hospital.” This is true, but the association of the park access is opposite—negative for binomial model and positive (although slightly) for SMR model. In addition, the model includes both physical inactivity and park access, so that the coefficient of the park access is interpreted as “conditioned on the same level of physical activity, park access would decrease mortality by ….” On the other hand, the authors interpret that park access could have beneficial health effects through promoting physical activity. This may be a little contradiction.
Author Response
We would like to thank the reviewer for their time and effort to provide us with valuable feedback on this paper. We have made changes to the manuscript to reflect the suggestions that have been provided. We hope our edits and responses below can address these concerns.
Point 1:
Thank you for taking the time to revise the manuscript and to consider my comments.
I understand the limitation as to data availability. I saw CDC’s website providing tabulated mortality data, but that may not be the case of the authors’ data.
Although the authors employed my second approach, I do not think they responded the first half of comment #1. It seems natural to me to model SMR, where the exponential of the coefficients represents the SMR ratio by the corresponding variables. In this approach, the estimated effects were not so strong, the reasonable finding that exemplifies the dilution of individual-level associations in ecological studies. In contrast, the main analysis of this study seems excessively complicated in that it models whether the naïve point estimates happened to fall below or above their median, which is itself a random variable specified by the point estimates. The results suggest that the latter approach appears to overestimate the effects of the variables compared with the SMR approach. Although I may be wrong, I suppose that the apparent overestimation may have resulted from picking up random variations because of the randomness of the dichotomic threshold (the random variable median).
Response 1:
Thank you for the comment. We agree with the reviewer that modeling the log SMR is a very sound approach. We also believe that modeling the dichotomized outcomes using binomial generalized linear mixed model is insightful even with its limitations as mentioned by the reviewer. Thus, we decided to include both analyses in the study. We agree with the reviewer’s comments regarding the limitations, and we have included the following in the limitation section –
“It should be noted that rates extracted from the source were estimated rates and their precision depends on the number of counts and population size of each county. Dichotomization of the outcomes in this study could distort the association of the variables with the outcomes. The estimated effects when modeling the log standardized hospitalization/mortality ratio were not strong compared to modeling the dichotomized outcomes.”
We do not believe that “finding that exemplifies the dilution of individual-level associations in ecological studies” is applicable in this case as both approaches used county-level data and individual-level data was not used in this study. However, the differences in the significance of association and strength of association between the two approaches could be explained by the thresholds used in the two approaches. For the hospitalization rate, the median hospitalization rate for dichotomization was 11.7 in this study whereas the hospitalization rate of the reference national population was 11.8 in the log standardized hospitalization ratio approach. For the death rate, median SDR in this study for dichotomization was 39.0 whereas the death rate of the reference national population was 37.4 in the log SMR model approach. This could explain why a few variables had a change in significance level and direction of association with the outcomes in the two approaches.
Point 2:
Followings are secondary comments—ignorable if you employ only SMR approach.
The authors say, “all variables which were significant to the multivariate binomial mixed model were also significant in the model for log standardized hospitalization ratio with the exception of age-adjusted obesity and presence of hospital.” This is true, but the association of the park access is opposite—negative for binomial model and positive (although slightly) for SMR model. In addition, the model includes both physical inactivity and park access, so that the coefficient of the park access is interpreted as “conditioned on the same level of physical activity, park access would decrease mortality by ….” On the other hand, the authors interpret that park access could have beneficial health effects through promoting physical activity. This may be a little contradiction.
Response 2:
Thank you for the comment. The difference in the direction of association of park access in the two models could be explained partially by the different threshold used to define the dichotomized outcome and the standardized mortality ratio as explained above. We have included the following in the limitations section:
“The differences the significance and strength of association of the variables with the outcomes in the two models can be explained partially by the difference in the threshold used in the two models.”
As for inclusion of both park access and physical inactivity, the correlation between the two was only -0.31. As for the contradiction in the discussion regarding this result, we have modified the discussion to suggestion park access are beneficial not only in promoting physical inactivity but also better environment and psychological health.